# Environmental Partitioning, Spatial Distribution, and Transport of Atmospheric Mercury (Hg) Originating from a Site of Former Chlor-Alkali Plant

Mert Guney [1,2], Aiganym Kumisbek [1,2], Zhanel Akimzhanova [1,2], Symbat Kismelyeva [1,2], Kamila Beisova [1,2], Almagul Zhakiyenova [1,2], Vassilis Inglezakis [3] and Ferhat Karaca [1,2,*]

[1] The Environment & Resource Efficiency Cluster (EREC), Nazarbayev University, Kabanbay Batyr Ave. 53, Nur-Sultan 010000, Kazakhstan; mert.guney@nu.edu.kz (M.G.); aiganym.kumisbek@nu.edu.kz (A.K.); zhanel.akimzhanova@nu.edu.kz (Z.A.); symbat.kismelyeva@nu.edu.kz (S.K.); kamila.beisova@nu.edu.kz (K.B.); almagul.zhakiyenova@nu.edu.kz (A.Z.)
[2] Department of Civil and Environmental Engineering, School of Engineering and Digital Sciences, Nazarbayev University, Kabanbay Batyr Ave. 53, Nur-Sultan 010000, Kazakhstan
[3] Chemical and Process Engineering, University of Strathclyde, Glasgow G1 1XQ, UK; vasileios.inglezakis@strath.ac.uk
* Correspondence: ferhat.karaca@nu.edu.kz

**Abstract:** Mercury (Hg) is one of the trace toxic and bioaccumulative global pollutants, and due to its long atmospheric lifetime, it presents a significant global challenge. The present study (1) utilizes total gaseous mercury (TGM) measurements made around a former Hg-cell chlor-alkali plant (CAP) located in Pavlodar, Kazakhstan, and predicts the spatial distribution of Hg over its premises and the nearby city. It then (2) estimates the environmental repartition of Hg deposited by the CAP using three fugacity models of varying complexity: Level I, QWASI, and HERMES. Finally, it (3) predicts long-range Hg transport via forward trajectory-based cluster analysis. The atmospheric Hg levels measured in Pavlodar and around Lake Balkyldak were elevated: in the range of 1–37 ng/m$^3$ with an urban background level at 4.9 ng/m$^3$. Specifically, concentrations up to 37 ng/m$^3$ close to Lake Balkyldak and up to 22 ng/m$^3$ nearby the city's industrial zone (where the CAP was located) had been observed. Interpolation maps created using kriging also suggest these locations as the primary sources of atmospheric Hg in the city. The Level I fugacity model indicated that almost all of Hg is expected to end up in the atmosphere. The modeling results obtained using more complex QWASI and HERMES models showed that some significant quantity of Hg would still be associated with the sediments of Lake Balkyldak (a large wastewater discharge pond nearby the CAP). The forward trajectory-based cluster analysis method revealed the long-range atmospheric transportation routes and local, regional, and global impact zones. Furthermore, a source-receptor relationship using air transportation pathways to identify "areas of impact" was addressed. During both heating and non-heating seasons, the frequency-based analysis identified the distribution of Hg reaching the territories of Mongolia, northwest China, southwest Kazakhstan. The Hybrid Single-Particle Lagrangian Integrated Trajectory (HYSPLIT-4) model forward trajectory analysis has confirmed similar patterns during heating and non-heating seasons, except with shorter impact distances during the non-heating period. Even though the CAP was closed more than 30 years ago and those past remediation efforts cleaned up the site, the residual Hg pollution seems significant and should be further investigated in different environmental media.

**Keywords:** air trajectory modeling; ArcGIS; contaminated site characterization; fugacity; HYSPLIT; kriging; Kazakhstan; Lake Balkyldak

## 1. Introduction

Atmospheric mercury (Hg) is one of the most dangerous global pollutants due to its atmospheric lifetime and toxicity [1–3]. There are three most critical Hg species in the

troposphere: gaseous elemental mercury ($Hg^0$), reactive gaseous or oxidized mercury, and total particulate mercury [4,5]. Atmospheric Hg emission sources can be natural (water/soil surface evasion, emission from earth's crust, volcanic eruptions, bush and forest fires) or anthropogenic (biomass and coal combustion, metal smelters and incinerators, gold mining, and other industrial processes); however, compared to the pre-industrial levels, anthropogenic activities have increased Hg emissions and cycling in the environment by a factor of three to five [3,6,7].

After being released to the atmosphere, elemental Hg can travel over long distances due to residence time up of to one year before its removal through oxidation on particle surfaces as well as gas-phase dry and wet deposition [8,9]. Once deposited in water or terrestrial bodies, it may be further oxidized into methylmercury (MeHg), which can easily bio-accumulate in the food chain and increase exposure to humans and wildlife [10]. Because of its long-range atmospheric transportation routes, Hg may be distributed globally and reach even remote areas [11]. Moreover, gaseous elemental Hg can be directly inhaled, and most dose exposures by inhalation are absorbed into the blood and quickly pass through the blood–brain barrier, crossing into the extracellular fluid of the central nervous system and as a result causing damage to the brain [12].

Chlor-alkali plants (CAPs) utilizing the Hg-cell method in their operations are among the most significant industrial Hg sources [13–17]. According to the U.N. Environment Program, the chlor-alkali industry is the third primary Hg user worldwide among industries [18]. During the CAP operation process, Hg is extensively used as a catalyst during brine water electrolysis to produce chlorine and sodium hydroxide and is often discharged with wastewater, solid waste, and atmospheric emissions. This type of plant may continuously release Hg to the surrounding environment (water, soil, and air) not only during the production process but also after its operations are seized [7,14,16,19,20]. Numerous studies have assessed the impact of Hg emissions resulting from the use of the Hg-cell technique in operations of CAPs worldwide [7,14,16,19,20]; some of these studies have focused on atmospheric Hg emissions [21–23]. Most of these assessments reported elevated Hg concentrations in multiple compartments violating local regulations and posing significant health risks to the population near the CAPs.

According to estimates, an old CAP (a former USSR military-industrial establishment), which operated in Pavlodar city in northern Kazakhstan between 1975–1993, discharged around 1500 g of Hg per ton of caustic soda produced [24,25]. An estimated loss of about 1310 tons of metallic Hg has been reported [25]. It can be stated that Pavlodar is one of the hotspots of Hg contamination with the potential to affect local areas as well as remote locations adversely. To the best of our knowledge, published research is limited regarding the local contamination in water and terrestrial systems [26–30] and environmental risk assessment issues on the contaminated site along with Hg bioremediation for soil treatment and site management [25,31–33], and no peer-reviewed study dealing with the atmospheric cycle and transportation dynamics exists. Further research is required on the atmospheric impact of the pollution on nearby and remote locations as well as on an up-to-date site assessment followed by human health risk characterization studies using the most up to date modern tools.

Recent literature addresses the need for the availability of new global datasets and global Hg cycling pathways covering areas of the world where environmental Hg data were previously lacking [12]. In addition to that, Hg monitoring research and CAP site characterization literature are not generally accompanied by ambient air measurements in the contaminated zone assessment campaigns [21,34–37]. In this way, an integrated approach for impact assessments of Hg-contaminated zones along with the measurement of ambient levels, local distribution characteristics, and long-range atmospheric transport pathways are required. Software for statistical analysis and mapping of the field measurements such as ArcGIS and its built-in tools including a geostatistical interpolation method called kriging are commonly used for air pollution [38], climatological studies [39], and soil mapping [40]. In addition to the general environmental assessment of the contaminated

sites, geostatistical analysis of data can aid in identification of the hotspots and, therefore, has been implemented in the present study.

Being able to predict the behavior of the contaminant once it is released to the environment, namely its fate and transport within different media (e.g., air, water, soils, sediments, and biota) and how long it may persist is a useful tool due to a possible long-range transport of certain toxic chemicals [41]. It is crucial to ensure that remediation actions are targeting the exact species in specific phases, and no risk is over- or underestimated. One of the available methods to predict a target chemical substance's spatial and temporal distribution is fugacity-based modeling [42]. Fugacity (*f*) describes the tendency of volatile chemicals to escape the medium and is essentially a mass-balance method based on the physicochemical properties of the contaminant and site-specific parameters [43]. Using the fugacity approach, one can estimate transport characteristics, concentrations, and bioaccumulation of the contaminant [42]. The term "aquivalence" or "aqueous equivalent concentration" introduced by Diamond et al. [44] is an equilibrium criterion analogous to fugacity, but suitable for modeling the environmental behavior of both volatile and non-volatile chemicals (e.g., metals, polymers, and certain polar compounds).

Since Hg is ubiquitous, non-degradable, and exists in multiple chemical species, it is necessary to effectively characterize both the spatial and temporal distribution of the contaminant in an evaluated environment. The majority of the Hg-specific fugacity/aquivalence models include three species–$Hg^0$, MeHg, and residual Hg (THg–$Hg^0$–MeHg, which almost completely consists of HgII); and simulate main interspecies transformation processes, such as $Hg^0$ oxidation, HgII reduction, HgII methylation, MeHg demethylation and degradation [45]. It should be noted that this type of contaminant fate and transport modeling has not yet been conducted for the present site before.

The objectives of this study are (1) to identify where and in which quantities the contaminant will end up in the environment using fugacity modeling, (2) to understand the baseline atmospheric ambient levels in the Hg-contaminated zone in Pavlodar and its distribution characteristics in the surrounding area; and, (3) to evaluate the impact of air masses originated from the contaminated site using a source-receptor relationship approach and identify the impact zones of its long-range atmospheric transport.

## 2. Materials and Methods

### 2.1. Study Area

The Hg pollution near Pavlodar (52°18′56″ N, 76°57′23″ E) city in north-eastern Kazakhstan is the result of the operation of the former P.O. "Khimprom" chlor-alkali plant (further referred to as CAP) with high chlorine and caustic soda production capacity of approximately 100,000 ton/y using Hg-cell technology in 1975–1993 [24]. Lake Balkyldak, which is located to the north from the CAP, is considered as one of the primary pollution sources as well because the major part of the Hg containing sludge and other industrial waste were released into such settling lagoons. Scientists, local environmental NGOs, regional and local authorities of Kazakhstan have persisted in reducing Hg contamination. After the first stage of the demercuration program in 2005, a comprehensive field study was completed with external financial support. However, due to limited investments, the initial demercuration plan was not fully completed [46]. Though the facility was shut down in 1992 with the current Pavlodar Chemical Plant (PCP) operating at its place and despite the remediation efforts undertaken, it can be still considered a Hg hotspot in the region (Figure 1), causing high risks of elemental dispersion into other media.

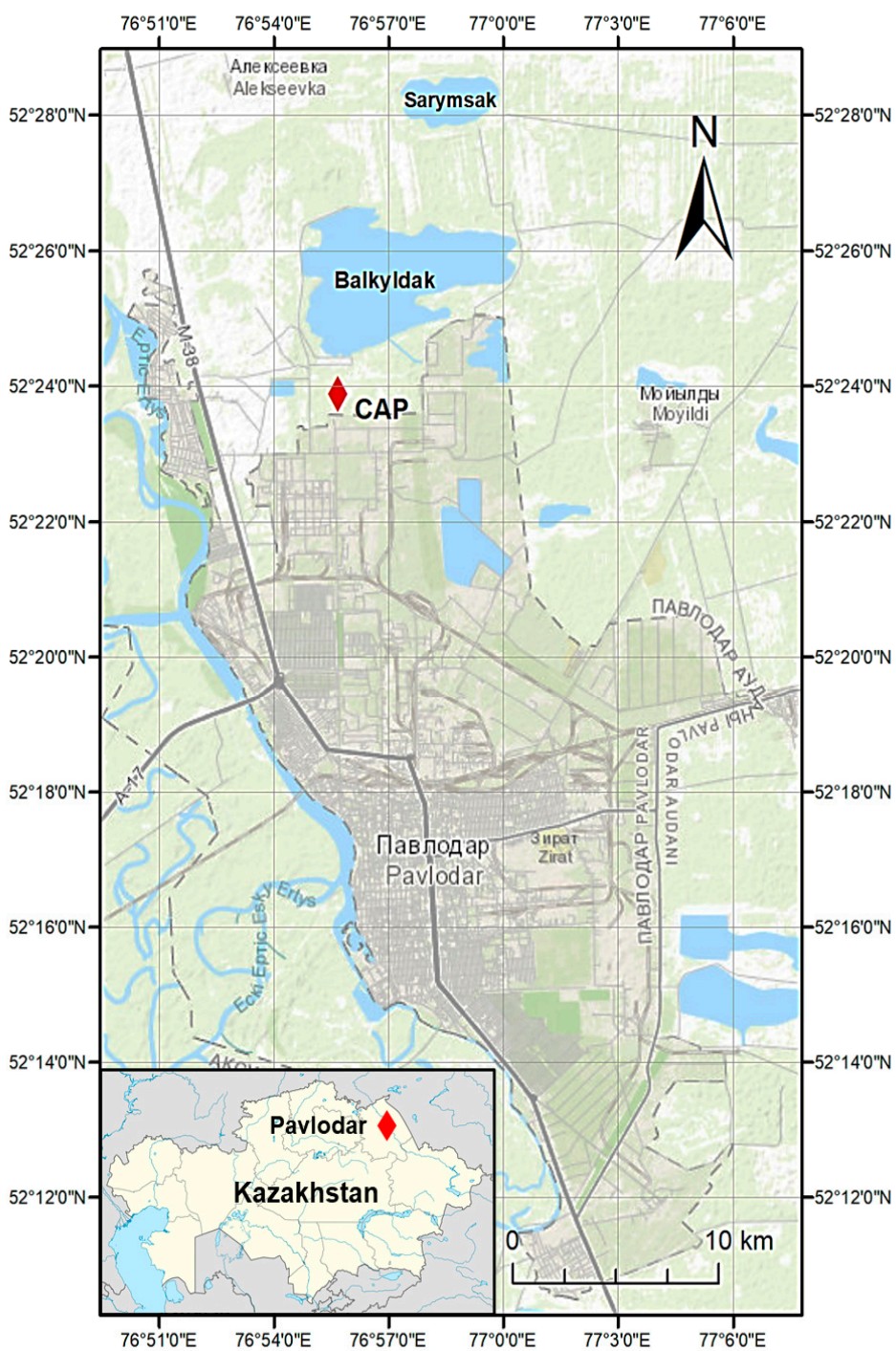

**Figure 1.** Location of study area.

*2.2. Spatial Distribution of Atmospheric Hg*

2.2.1. Atmospheric Hg Measurements

A portable Hg analyzer (RA-915 Portable Zeeman Hg Analyzer) was used in direct ambient air Hg measurements, operating as an atomic absorption spectrometer at 254 nm with Zeeman correction for background absorption for interference-free measurement with a low detection limit (0.5 ng/m$^3$) and a wide measurement range. The quality control procedure of the instrument involved the equipment's internal calibration check performed before each campaign where the instrument runs a standard algorithm and provides a relative deviation of total gaseous mercury (TGM) concentration measured in the test cell from its expected signal value (below 20% is acceptable). The instrument

has also been performing periodic zero checks set at every 10 min using an internal calibration cell. Several ambient measurements using both random and systematic grid sampling approaches were performed over the selected study area during four sampling campaigns. Figure 2 illustrates the sampling points of the campaigns for Hg measurements on consecutive days for 23–26 July 2019. The meteorological conditions during that period were generally similar with a cloudy and slightly rainy weather, air temperature between +16 and +23 °C, and light wind from north and west at 2.4–3.7 m/s (except for Campaign 1 with 0.1–1.1 m/s) resulting in similar atmospheric dispersion characteristics. Ambient Hg levels over rural areas, including the surroundings of Lake Balkyldak, were measured during the first and second campaigns; whereas the focus of the third and fourth campaigns were daytime and nighttime atmospheric Hg levels in the city, respectively.

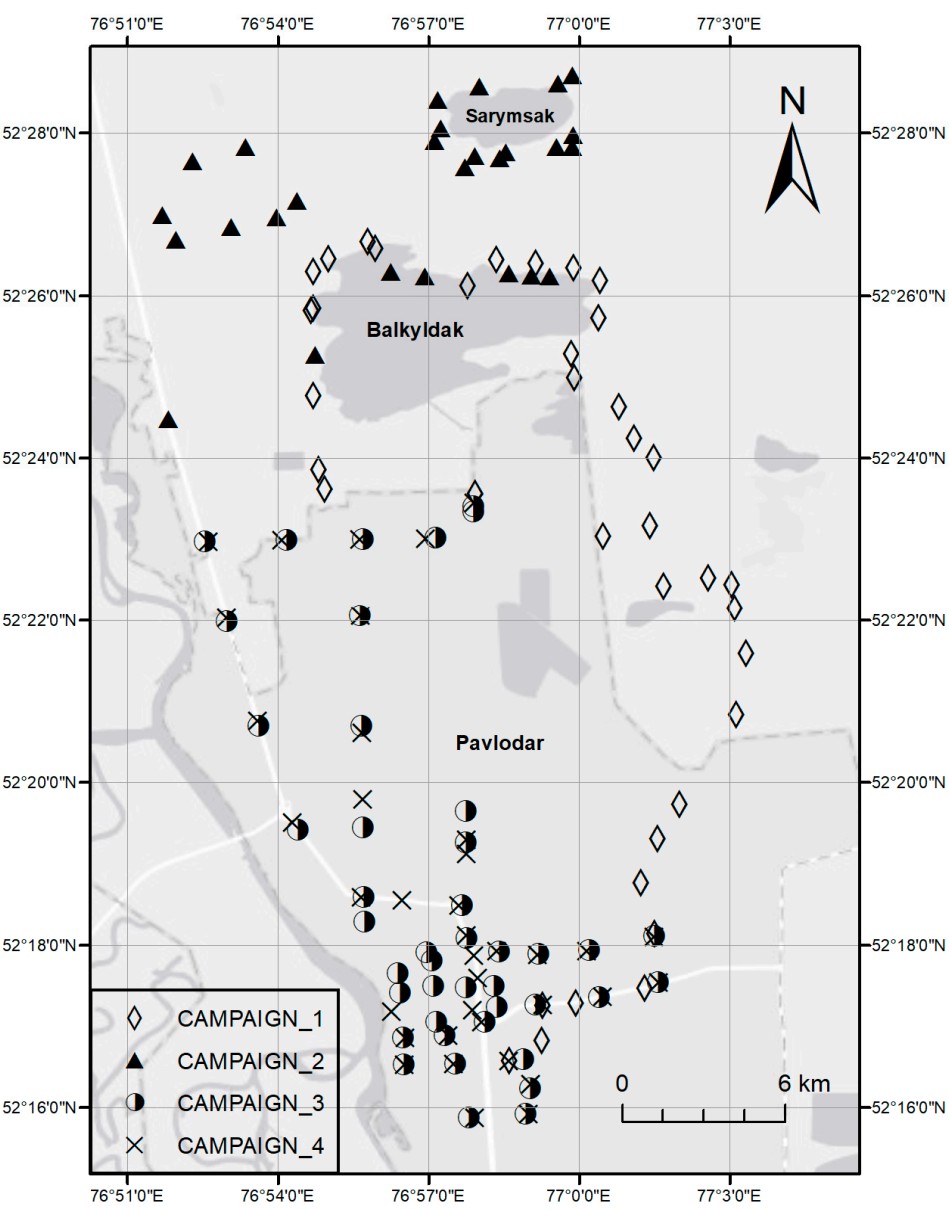

**Figure 2.** Sampling points of four Hg measurement campaigns: (1) July 23 (daytime), (2) July 24 (daytime), (3) July 25 (nighttime), (4) July 26 (daytime).

### 2.2.2. Geostatistical Analysis and Mapping

Measured Hg concentrations and the corresponding geographical coordinates (obtained using Garmin GPSMAP 64s) have been used to produce spatial Hg distribution maps (ArcGIS 10.6). Though kriging is able to achieve the least biased prediction results regardless of data distribution, the accuracy of prediction increases with closer resemblance of normally distributed data [40]. Therefore, the Hg concentration data were first checked for normality using the Shapiro–Wilk test and visual inspection of histograms and quantile-quantile (Q.-Q.) plots and then normalized to the best extent using ArcGIS. The statistical analyses have revealed significant skew in all four non-transformed datasets which is a common trait among environmental datasets, and high kurtosis (heavy tails or outliers) in Campaigns 3 and 4. Subsequently, built-in extensions called Geostatistical Analyst and Geostatistical Wizard were used to produce semivariograms and interpolation maps using the ordinary kriging method. A log transformation of Campaign 1 data achieved normal distribution. It also substantially improved the normality by reducing skewness and kurtosis for Campaign 2 and 3 measurements. Campaigns 2–4 could not be fully normalized through common transformation methods mainly due to a small range in the measured concentrations. Finally, the cross-validation of interpolation results was performed.

### 2.3. Atmospheric Hg Transport Model

### 2.3.1. Fugacity Model for Hg Transport

The fundamental relationship used in fugacity estimations is the linear relationship of the contaminant's concentration in a certain compartment ($C_i$, mol/m$^3$) to fugacity ($f$, Pa) and the compartment's fugacity capacity ($Z_i$, mol/m$^3$/Pa):

$$C_i = Z_i \times f \tag{1}$$

In the system, where compartments are in equilibrium, fugacity is equal, which enables one to calculate relative concentrations of the chemical substance in different media by first estimating their fugacity capacities. The higher the compartment's fugacity capacity—the more of the contaminant it can "retain" [47]. With a handful of fugacity-based models available today and several studies applying the fugacity approach to Hg in environmental systems, especially in lakes [43,44], the main challenge becomes a determination of the environmental parameters and data on contaminant releases. Tables 1–3 present the site-specific environmental parameters of the Pavlodar region as well as Hg properties used in the modeling as well as model results.

**Table 1.** Environmental parameters and Hg properties used in fugacity estimations.

| Hg$^0$ | Value | Source |
|---|---|---|
| Data Temp ($^\circ$C) | 25 | Assumed |
| Molecular weight (g/mol) | 200.59 | HERMES model |
| Log K$_{OW}$ | 0.623 | HERMES model |
| Melting point ($^\circ$C) | $-38.87$ | HERMES model |
| Water solubility (g/m$^3$) | 0.0334 | HERMES model |
| Lake area (km$^2$) | 15 | [26] |
| Lake mean depth (m) | 4.5 | [26] |
| Lake volume (m$^3$) | 67,500,000 | Calculated |
| Sediment active layer depth (m) | 0.01 | [41] |
| Rain rate (m/yr) | 0.25 | [26] |

**Table 1.** *Cont.*

| Hg$^0$ | Value | Source |
|---|---|---|
| ρ Air (kg/m$^3$) | 1.185413 | Widely accepted value |
| ρ Aerosol (kg/m$^3$) | 2000 | [41] |
| ρ Water (kg/m$^3$) | 1000 | Widely accepted value |
| ρ SPM (kg/m$^3$) | 1500 | [41] |
| ρ Fish (kg/m$^3$) | 1000 | [41] |
| ρ Soil (kg/m$^3$) | 2400 | [41] |
| ρ Sediments (kg/m$^3$) | 2400 | [41] |
| OC fraction in SPM (g/g) | 0.2 | [41] |
| OC fraction in soil (g/g) | 0.02 | [41] |
| OC fraction in sediment (g/g) | 0.04 | [41] |
| Fish lipid content | 0.03 | Selected based on fish types mentioned in [48] |
| Total mass discharged (kg) | $1.00 \times 10^6$ | [26,49] |
| Estimated Hg mass discharged in the lake (kg) | 135,400 | [25] |

**Table 2.** Hg concentrations in different media as predicted by fugacity models.

| Model (Total Mass in the System) | Level I (1000 t) | | QWASI (135.4 t) | | HERMES (135.4 t) |
|---|---|---|---|---|---|
| | Mass | Concentration | Mass | Concentration | Concentration |
| Air (kg, ng/m$^3$) | $9.98 \times 10^5$ | $9.98 \times 10^6$ | - | $1.70 \times 10^1$ | $1.70 \times 10^1$ |
| Aerosol (kg, ng/m$^3$) | $4.96 \times 10^2$ | $4.96 \times 10^3$ | - | - | - |
| Soil (kg, ng/g) | $1.90 \times 10^1$ | $5.87 \times 10^{-1}$ | - | - | |
| Water (kg, ng/L) | $1.15 \times 10^3$ | $1.71 \times 10^4$ | $3.54 \times 10^3$ | $5.24 \times 10^4$ | $1.05 \times 10^6$ |
| Suspended sediment (kg, ng/g) | $2.76 \times 10^{-4}$ | $5.87 \times 10^0$ | $5.64 \times 10^{-4}$ | $1.80 \times 10^1$ | $2.04 \times 10^2$ |
| Sediment (kg, ng/g) | $4.23 \times 10^{-1}$ | $1.17 \times 10^0$ | $1.17 \times 10^{-1}$ | $3.61 \times 10^0$ | $8.29 \times 10^6$ |
| Fish (kg, ng/g) | $1.45 \times 10^{-4}$ | $2.15 \times 10^0$ | - | - | - |

**Table 3.** HERMES results by Hg species: elemental (Hg0), methylmercury (MeHg), residual (ResHg).

| | Dissolved Hg$^0$ (ng/L) | Dissolved MeHg (ng/L) | Dissolved Res Hg (ng/L) | Solids Hg$^0$ (ng/g) | Solids MeHg (ng/g) | Solids Res Hg (ng/g) |
|---|---|---|---|---|---|---|
| Air (kg, ng/m$^3$) | $1.67 \times 10^1$ | $8.33 \times 10^{-2}$ | $2.50 \times 10^{-1}$ | - | - | - |
| Water (kg, ng/L) | $2.09 \times 10^4$ | $3.09 \times 10^4$ | $1.00 \times 10^6$ | $4.04 \times 10^0$ | $5.98 \times 10^0$ | $1.94 \times 10^2$ |
| Sediment (kg, ng/g) | $1.07 \times 10^6$ | $2.14 \times 10^4$ | $3.42 \times 10^6$ | $2.19 \times 10^7$ | $4.38 \times 10^5$ | $6.97 \times 10^7$ |

In the present study, three different fugacity/aquivalence models have been used, such as Level I model—a closed mass-balance steady-state equilibrium (no reaction) model with a defined amount of input chemical that partitions in seven compartments (air, aerosol, water, soil, sediment, suspended particulate matter (SPM), fish). Traditionally, aerosols are not considered in the Level I model [41]; however, non-zero volume input was required for aerosol by the software. The second model, the Quantitative Water Air Sediment Interaction (QWASI) model for lakes, is used for estimating chemical concentrations and fluxes (emissions, advection, transport between media) for three compartments, i.e., sediments,

water, air [50]. The third model, Hg Environmental Ratios Multimedia Ecosystem Sources (HERMES), is also a mass balance steady-state model like QWASI. Still, unlike the latter, it considers three main Hg forms ($Hg^0$, MeHg, and residual Hg—mainly consisting of HgII) and dynamic transformations between them. The HERMES model has two segments—a water column and sediment [51].

Elemental $Hg^0$ is the "key species" that is used in the non-Hg-specific models (Level I and QWASI) since it is present ubiquitously in the environment, at least in small quantities, and is not readily reactive with many chemicals [51]. In Level I and QWASI models, the distribution into the compartments for only the elemental form of Hg was estimated.

2.3.2. Trajectory Modeling for Hg Transport in the Atmosphere

Forward trajectories, along with clustering and frequency analysis methods, are used to evaluate the spatial characteristics of potential long-range Hg transportation patterns originating from Pavlodar. Since pollutants enrich the air parcels (e.g., elemental Hg), they could affect possible downwind locations by deposition processes during their transport. The suggested approach helps to determine the most common air transportation routes, which are the most likely "impact zones" in terms of trajectory endpoints frequencies and their distribution distances on the global map [52–58].

The Hybrid Single-Particle Lagrangian Integrated Trajectory Model (HYSPLIT) is a trajectory model developed by The National Oceanic and Atmospheric Administration (NOAA) Air Resources Laboratory (ARL) [59,60]. This model is one of the widespread trajectory modeling systems, where air parcel trajectories, distributions, and most likely transport destinations can be computed [61]. Three days air parcel forward trajectories originating from the Pavlodar Hg-contaminated site were computed every hour for a year (e.g., 2017). Starting an average ground level of 100 m was set as the starting elevation for the hourly runs. The Global Data Assimilation System (GDAS-1, one-degree archive, available at ftp://arlftp.arlhq.noaa.gov/pub/archives/gdas1) meteorological database was employed in the trajectory calculation stage. While the trajectory clusters indicate the most likely routes of air parcel trajectories, the cumulative frequency maps of the calculated trajectories represent the areas of impact with potential probabilities.

## 3. Results and Discussion

### 3.1. Spatial Distribution

Four Hg monitoring campaigns have been carried out in July 2019 with measured Hg concentrations ranging from 1 to 37 ng/m$^3$ with an average value of 7.5 ± 6.3 ng/m$^3$. The urban background level of Hg was determined as 4.9 ng/m$^3$, as an average value of Hg concentrations measured at least 5 km away from the source (in the present case, the CAP and Lake Balkyldak). Geospatial interpolation-based visualized atmospheric Hg levels recorded during the measurement campaigns in Pavlodar and around Lake Balkyldak and its surrounding area are given in Figures 3 and 4.

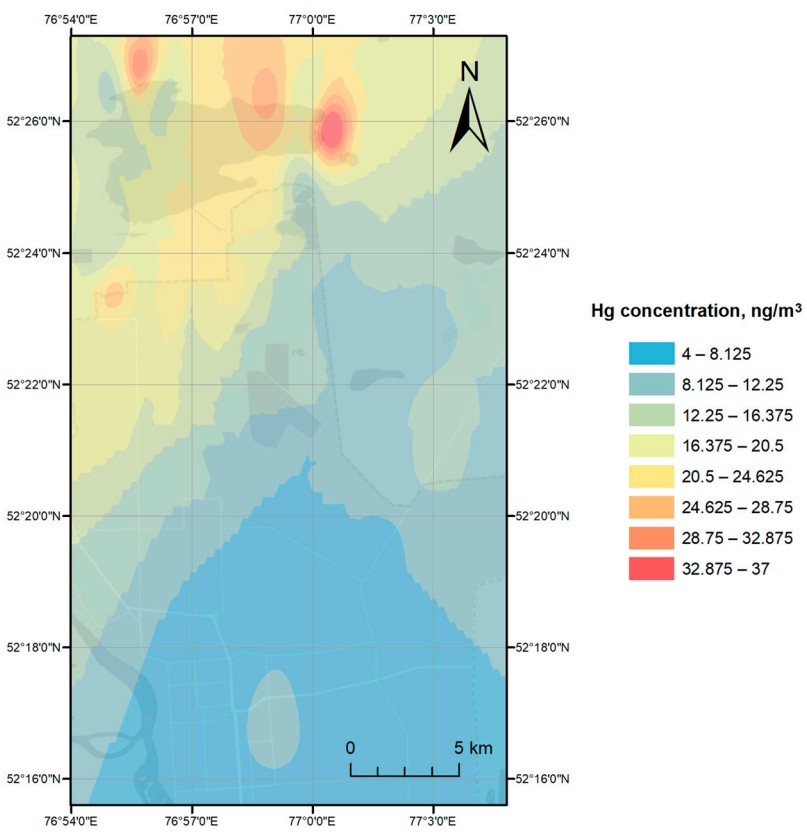

**Figure 3.** Atmospheric Hg dispersion over region around Lake Balkyldak during Campaign 1.

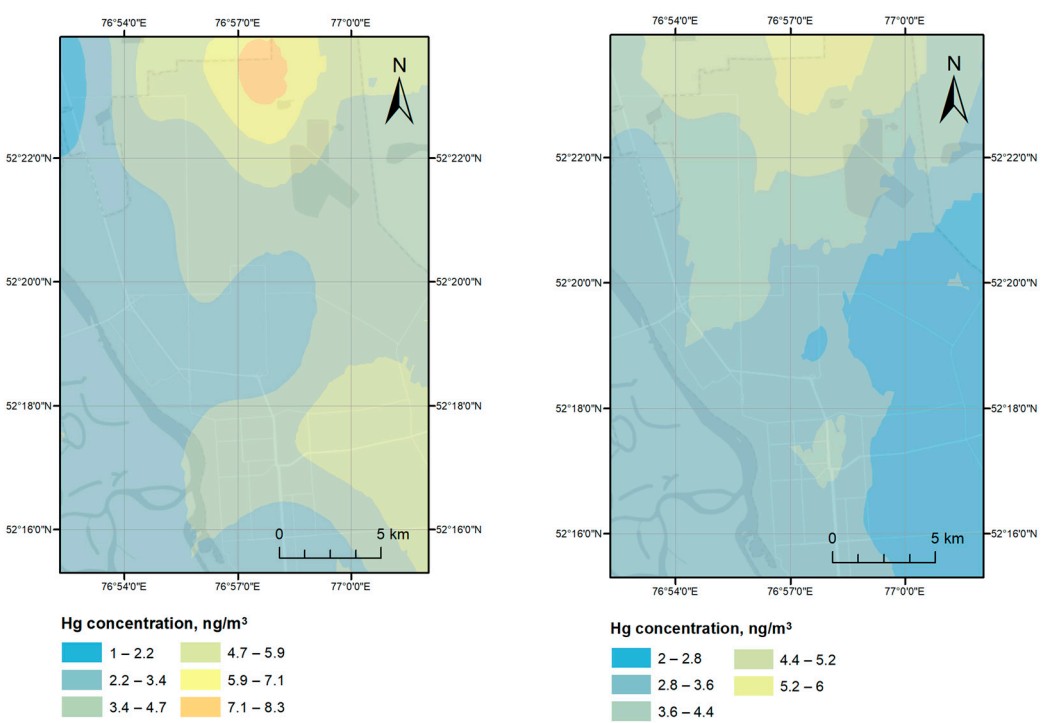

**Figure 4.** Atmospheric Hg dispersion over study area during Campaigns 3 and 4.

The spatial distribution presented may be affected by certain factors. First, the sampling plan was subject to topographical and site access limitations which may lead to an underrepresentation of the expected higher Hg pollution around the CAP. Second, the

spatial distribution is dependent on the wind patterns, thus the temporal Hg distribution will vary to different extents under changing atmospheric conditions.

### 3.1.1. Interpolation

Spatial correlation of the dataset is crucial to geostatistical interpolation: measurements at points in close proximity should have similar values. The spatial correlation of the field data can be verified using semivariograms, where the semivariance of each pair of sampled points is plotted against the distance between the points. Semivariograms for each measurement campaign followed the natural path with low semivariance for closer points, sharp increases, and gradual leveling out at more considerable distances. Table S1 (see Supplementary Material) lists the major parameters of the generated semivariograms. The field measurements during four campaigns have been provided in Table S2.

Randomized measurements of atmospheric Hg levels, around Lake Balkyldak as well as its surroundings, was conducted on 23 July 2020 (11:30 to 17:00) (Campaign 1, Figure 3). The key meteorological conditions in that time frame were as follows: fog and light rain in the morning and cloudy weather in the afternoon; air temperature: +16 to +22 °C; wind speed and direction: 0.1 to 1.1 m/s from west. The overall range of recorded Hg concentrations for the campaign was 4–37 ng/m$^3$, with the highest levels aggregated around northwestern (24–30 ng/m$^3$) and north-eastern (18–37 ng/m$^3$) shores of the lake, a modest peak (16–22 ng/m$^3$) in the industrial zone between the city and Lake Balkyldak, and a lower concentration range (4–10 ng/m$^3$) in the city. Aggregation of points with extremely high Hg concentrations around Lake Balkyldak could indicate its strength as a Hg source and may be supported by low to no air movement. The field measurements of Campaign 2 on 24 July 2020 (10:00 to 16:00) were limited to the lakes at the north of the city, including Balkyldak and Sarymsak, and yielded a lower Hg level range (6–9 ng/m$^3$). The average temperature and wind speed during the sampling hours were +18 to +20 °C and 2.5 (from the west) to 3.7 m/s (from the northwest), respectively, with foggy and rainy weather similar to the previous sampling day. Distribution of Hg levels in Campaign 2 (Figure S1, see Supplementary Material) within the sampling area seemed to be somewhat homogeneous with the exceptions to the north-eastern part of Lake Balkyldak with the lowest recorded concentrations (6 ng/m$^3$) and the western part of the entire area having marginally higher Hg levels (8–9 ng/m$^3$) in general. A decrease in the contaminant levels during Campaign 2 compared to Campaign 1 could be attributed to a higher wind speed during Campaign 2 measurements.

Field measurements in Campaign 3 were performed from approximately 21:30 to 00:00 to study the nocturnal atmospheric Hg concentrations over the city. Higher concentrations (6–12 ng/m$^3$) were recorded in the northern industrial district of Pavlodar city, which contains chemical plants, an industrial cluster manufacturing aluminum products, a metalworking factory, an oil refinery factory, etc., whereas lower Hg levels were detected in the northwestern, western, and southern parts of the city (1–3 ng/m$^3$). Industries manufacturing ferrous and non-ferrous metal products as well as oil refineries are also potential Hg sources due to "unintentional" Hg releases from combusting and processing Hg-containing fuels and ores [62]. Though the Hg emissions from these industries (0.05–0.5 g of Hg per t of coal burned, 0.04–0.7 g of Hg per t of Al produced, 0.01–0.5 g of Hg per t of pig iron produced, and 0.0034 g of Hg per t of crude oil refined) might augment to an overall increase in ambient Hg levels, these releases are marginal in comparison to emissions from CAPs utilizing Hg-cell technology (20 g of Hg/t of Cl$_2$ capacity) [63] and the legacy Hg releases from the plant's territory located next to the northern industrial district. The average temperature and wind speed for the last 6 h of the day were +23 °C and 2.4 m/s (from the northwest), respectively. Interpolation results in Figure 4 suggest that the source of airborne Hg is located north of the city in proximity to the former CAP territory with the wind carrying atmospheric Hg in the southeastern direction.

Campaign 4 measurements taken the next day between around 11:30 to 15:30 represent diurnal atmospheric Hg levels over the city with the range twice smaller than those during

Campaign 3 (2–6 ng/m$^3$), and thus, producing a similar but smaller concentration gradient from north to south. The daytime average temperature and wind speed between 12:00 and 18:00 were +20 °C and 3.0 m/s (from the north) on the sampling day, mirroring the Hg dispersion pattern of Campaign 3 seemingly affected by wind direction. Though mean Hg concentrations in the more densely populated part of Pavlodar ranged 2–5 ng/m$^3$ on both sampling days, there is a larger discrepancy between the measurements in the northern industrial district (nighttime: 6–12 ng/m$^3$, daytime: 5–6 ng/m$^3$).

### 3.1.2. Cross-Validation, Comparison to Literature

A cross-validation is performed to assess the model's accuracy in predicting values of the unsampled locations by using the rest of the dataset to predict the known values one by one and estimating prediction error based on the deviation of predicted values from the known ones [38]. For cross-validation of generated interpolation maps, the following parameters were used: regression function, root-mean-square-standardized, and average standard error (Table S1). The results were particularly satisfactory for Campaign 1 based on the obtained regression function and standardized error values. The data from the remaining campaigns, which had a flatter pattern, resulted in more mediocre cross-validation results.

Several studies assessed the environmental impact of atmospheric Hg pollution on the local area from Hg-cell chlor-alkali plants as the main source of Hg [14,37,64,65]. Atmospheric Hg concentrations measured outside of the facility's territory in China (1.35–9.80 ng/m$^3$, [14]) are lower and in Sweden (1.4–40 ng/m$^3$, [21]) are similar to those reported in the present study (1–37 ng/m$^3$); however, Hg levels reported for CAPs in Italy (2.8–100 ng/m$^3$, [64]) and Spain (3.5–229 ng/m$^3$, [37]) exceed those in this study considerably. Though the sampling locations in the present study have been selected such that they consisted of points in sensitive zones (i.e., urban areas) and in zones impacted by the former Hg releases, it should be noted that most of these points are not close to the CAP site in the down-wind direction. Moreover, studies by [14,65] measured significantly elevated Hg concentrations in the air of the plants' territories (1.5–540 in Sweden [65] and 1.40–1670 ng/m$^3$ in China [14]), suggesting that atmospheric Hg levels inside the former CAP territory in Pavlodar can exceed the Hg levels reported in the present study and should also be measured.

### 3.2. Fugacity Modeling

Regarding the fugacity modeling, input variables (Table 1) for three models have several common parameters. That being said, the more advanced the model is, the more detailed site-specific data is required; whereas in simpler models, the majority of the parameters are either assumed by default or are not considered at all. A detailed explanation of the parameters employed in each model is provided in Section S1 of the Supplementary Material (Tables S3–S5).

The Level I simulation (using the total estimated Hg loss from the site as input: 1000 t) indicates that 99.8% of the Hg eventually goes to the atmosphere, with a calculated concentration of 9,980,000 ng/m$^3$. In QWASI (using Hg input to Lake Balkyldak only: 135.4 t) the compartment with the highest concentration was water (52,399 ng/L); whereas in HERMES (using the same input as QWASI) max THg concentration was found in sediment compartment—8,286,268 ng/g. All of the three models are screening-level, not time-varying models; media in those models are assumed to be well-mixed and homogeneous, and all contaminant inputs occur simultaneously. The results are different in all models since the assumptions about the environment and levels of complexity are dissimilar. It is necessary to note that in HERMES, the results are shown for three Hg forms (Hg$^0$, MeHg, residual Hg), while in Level I and QWASI, the environments were simulated only for the discharge of elemental Hg. These results indicate that, among the two main types of emissions from the CAP, elemental Hg emissions ended up mostly in the air whereas discharges to Lake Balkyldak remained there in the water and in sediments.

In QWASI and HERMES models, the input value of Hg concentration in the air (measured in 2019—17 ng/m$^3$) is assumed as a final concentration, and the calculations of the concentrations in other compartments are performed accordingly—so it was predicted that the majority of the pollutant would be buried in the sediment; while in Level I, the value of Hg in the air is not considered (not demanded by the model). This might be a potential limitation for the results because as the plant ceased its activities more than three decades ago, the measurements of Hg content in the air from 2019 is definitely expected to show significantly lower values than the real ones from the 1990s. As a result, the calculated concentrations in QWASI and HERMES in other compartments might have been different if data from the 1990s would be available. More specifically, if higher expected air Hg concentrations in the past would have been available and then used in modeling, these would result in even higher concentrations of Hg in water and sediments (as supported by the sensitivity analysis presented below).

In the Level I model, the sensitivity analysis showed the only difference in total Hg mass is in suspended sediment: in the main model, with an SPM concentration of 0.463 mg/L, total Hg mass in the sediments was calculated to be 0.276 g; whereas it was ten times higher in the modified version with SPM of 5 mg/L. In the QWASI model, additional runs were done with three modified parameters, i.e., (1) with increased air Hg concentration, (2) with higher SPM, and 3) reduced inflow rate. In the first case, the final Hg concentration in the air was 300 ng/m$^3$ (equal to the input), and the fugacity changed from 0.21 μPa to 3.71 μPa. With SPM increased to 5 mg/L, the mass of suspended sediment also increased up to 6.09 g compared to 0.564 g with Hg in SPM of 0.463 mg/L. Modification of the water inflow rate from 22.5 to 0.001 m$^3$/h resulted in an insignificant effect: the growth of Hg in water content from 52,399 to 52,403 ng/L. Similarly, in the HERMES software, three extra models were run with altered parameters—with air Hg concentration of 300 ng/m$^3$ instead of 17 ng/m$^3$; with SPM of 5 mg/L instead of 0.463 mg/L; and river inflow changed from 22.5 m$^3$/h to 0.001 m$^3$/h. In the model with altered air concentration, it was noticed that the concentrations in all other media were almost not affected. In the model with the SPM concentration of 5 mg/L, the Hg concentration in water and in suspended solids increased (from 203 to 915 ng/g); and, on the contrary, the concentration in sediments decreased compared to the model with SPM of 0.463 mg/L. Reduced water inflow rate (from 22.5 to 0.001 m$^3$/h) did not considerably affect the concentrations in all media, but the advection out of the lake dropped from 268 to 0.0119 kg/yr.

The main limitation of fugacity calculations is that they usually make a simplification by assuming the equilibrium between all phases of the target chemical while ignoring transfer resistances between those phases [66]. Since HERMES is the only Hg-specific model among three, it provides the output in terms of three Hg species, which is of great importance since the toxicities and chemical properties differ widely for all forms of the element, and those are distributed in all media in inequivalent proportions. Despite the higher complexity of HERMES compared to Level I and QWASI, it is only a screening tool. Other more advanced models for fugacity modeling are available, and many of them are described in more detail in the literature [46]. If in-lake concentration gradients at different depths are important, a more advanced model may be recommended. On the contrary, simplified representations can be applied for lakes and screening-level risk assessment where one-dimensional (1D) modeling is sufficient and/or if there is a deficit of information about water and sediment transport in the system [51]. HERMES was compared to more complex mechanistic models (LOTOX2-Hg and simplified STELLA Hg), and the output from HERMES was well-matched with the actual measured values from lake Ontario or LOTOX2-Hg modeled values (r$^2$ = 0.955; $p$ < 0.0001; n = 10) [51].

A more improved field data used in conjunction with fugacity modeling may provide an effective site management tool for the present site. As an example, Japan created an air monitoring program that detects the concentrations of Hg and other heavy metals and tries to understand their long-range transport [67]. Such data may be used to reduce the health risk from exposure to different forms of Hg. If the quantities of Hg in ecosystem

compartments could be accurately predicted, this may further help improve the "big picture" and further assist with selecting effective remediation options.

*3.3. Trajectory Analysis and Interpretation*

Similar to the spatial distribution, the trajectory analysis has limitations as it may be affected by certain factors. First, the sampling plan was subject to topographical and site access limitations which may lead to an underrepresentation of the higher Hg pollution from the CAP area. Second, the Hg emission could be seasonally dependent due to changes in ambient temperature, thus the real trajectories would vary to different extents under changing atmospheric conditions.

3.3.1. Identification of the Areas of Impact

The frequency analysis of air parcel trajectories provides information about specific regions named "impact zones", which are the locations most frequently influenced by the air parcels originating from the pollution source area. Exposure to air pollutants is considered in terms of distance: local (0.0–0.2 km), regional (0.2–200 km), and global impact zone (200–2000 km) [68]. Air pollutants can spread into regions at detectable levels along the air parcels trajectory paths when the air parcels are transported in a sufficiently short time. Taking into consideration the average wind speed in the Pavlodar region (10 mi/h (~16 km/h)), the travel time of trajectories could be estimated [69]. The shorter time (<2 h) trajectories of higher frequencies enriched by atmospheric Hg are likely to present a higher impact on the local respirable air. When the trajectories transport at higher altitudes for longer times, they are expected to contribute more to the global Hg cycle. Thus, both trajectory frequencies (shorter periods e.g., 2 h or more extended periods e.g., 72 h) provide valuable information on local or long-range distances depending on the research interest. In the present study, the classification of zones of impact is done based on four trajectory frequency classes (%): direct impact zone with <100% of trajectory frequency, local impact zone with <10%, regional impact zone with <1%, and global impact zone with <0.1%.

The seasonal trajectory frequency maps in Figure 5 present the sum of the normalized frequencies of all the trajectories, originating from the former CAP site at the mid-atmospheric boundary layer (ABL, ~100 m above the ground level), for the grid cell locations where the trajectories passed over. The normalization was done by the total number of endpoints. A trajectory may intersect a grid cell once or multiple times based on its residence time in the grid. The results are discussed considering heating (mid of September–mid of April) and non-heating seasons (mid of April–mid of September). In general, the impact zones during non-heating and heating seasons suggest that the Hg contamination risks from local to global scales show similar patterns. For example, they typically extend more frequently over the north-eastern directions of the study area, and their travel altitudes in both cases increase and pass over the atmospheric boundary layer as the frequency areas reach long distances (Figure 5).

The area of impact with 0.1% frequency (the global impact zone) does not indicate a significant hazard level for the local and regional scales, and yet it is essential to understand the potential contribution of Hg released from the site to the global Hg cycle. This impact zone is likely to carry only trace amounts of Hg due to higher pollution dispersion time. The heating season plot has a higher impact directed to the north-east from the Pavlodar region, reaching up to the Laptev Sea (3000 km). The global impact area is dispersed over the whole territory of Mongolia, northwest of China, and southwest Kazakhstan. Meanwhile, in the non-heating season, it reaches Irkutsk, 2000 km away from the source, as well as reaching the southeast of Kazakhstan.

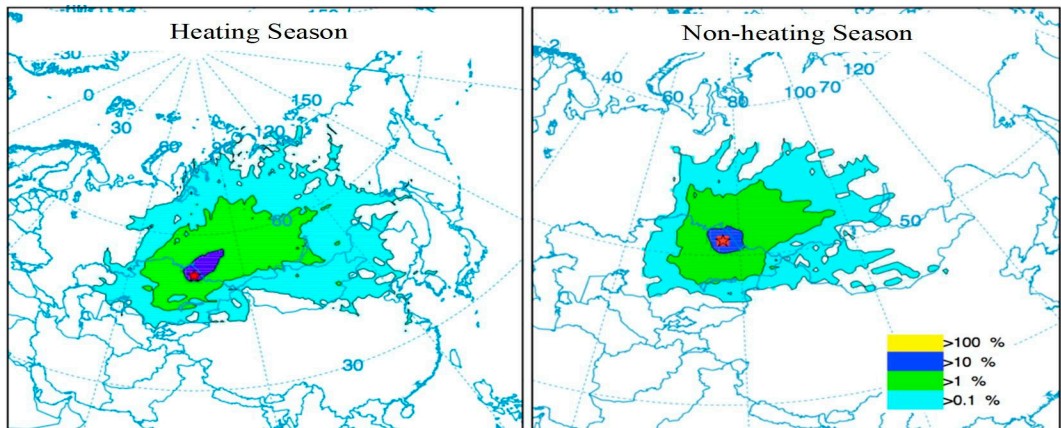

**Figure 5.** Most frequently influenced locations (areas of impact) by air parcels originated from region of pollution source.

The regional area of impact with the frequencies of less than 1% level points out the regional environmental risks (e.g., to increase trace amounts of methylmercury (MeHg) in the region). In this range, Hg impact on the air could be in detectable levels (e.g., $0.5 \ \text{ng/m}^3$) during downwind conditions since this area of impact is typically exposed to the trajectories traveling under the boundary level less than 12 h travel time. The regional impact zone extends over the north-east direction reaching Krasnoyarsk (1500 km), south and central Kazakhstan, and Lake Balkhash (1000 km). The frequency profiles of both seasons are quite similar.

The local impact zone with <10% frequency of trajectories is the zone indicating most significant environmental risks. It can be speculated that increased atmospheric Hg levels can be detected in particular locations in this zone during downwind conditions. However, the most significant impact can be attributed to its unremitting deposition in the region. Accordingly, the Hg emitted into the air from the Pavlodar region mostly settles into water or onto land as a result of wet and dry deposition over the area; then, it can be washed into the regional water systems. Once atmospheric Hg is deposited in the region, certain microorganisms can change it into MeHg, which is accumulated in the food chain, mainly fish and animals that eat fish [10]. Human exposure to Hg from eating the fish containing MeHg should be considered as a significant environmental risk for the population living in the region. Moreover, elevated Hg levels in the range of 11–43 $\text{ng/m}^3$ were reported in Novosibirsk and Krasnoyarsk, which are Russian cities in the impact zone. However, their elevated levels were attributed to the local industries, mainly gold refining plants [70] and an aluminum factory in Krasnoyarsk [71]. Along with operating anthropogenic sources of atmospheric Hg in the mentioned region, the Central Asian region, including Siberia, is located within one of three planetary mercury-ferrous belts [72]. On the other hand, [73] suggested that Hg-contaminated water from the Irtysh river in the Pavlodar region might be causing pollution in the territories of the impact area. It can still be speculated that some part of atmospheric Hg detected in these cities could be attributed to the Hg emissions from Pavlodar if the atmospheric Hg levels rise in the global and regional impact areas when they are located downwind of Pavlodar due to the possible presence of trace amounts of Hg in the atmosphere at detectable levels.

### 3.3.2. Atmospheric Transportation Routes

As the study focuses on the atmospheric Hg transport and distribution from a contaminated area, the performed analysis is considered only employing forward trajectories. HYSPLIT-4 cluster analysis (CA) is used to identify and analyze multiple trajectories of Hg flow patterns, the main transport pathways for the heating and non-heating seasons. Using CA, it was possible to divide trajectories into representative groups considering their shapes and characteristics. The purpose of CA was to identify the main paths of Hg transportation from its source to other regions. In order to identify the optimum number of

clusters, the total spatial variance measure (TSV, %), was compared against the number of clusters extracted (Figure 6). When the number of clusters was increased, the TSV showed a monotonic decreasing behavior with insignificant variations. However, there is a noticeable jump among points between 1 and 6 for all three plots. So the optimum number for the clusters was chosen to be 6.

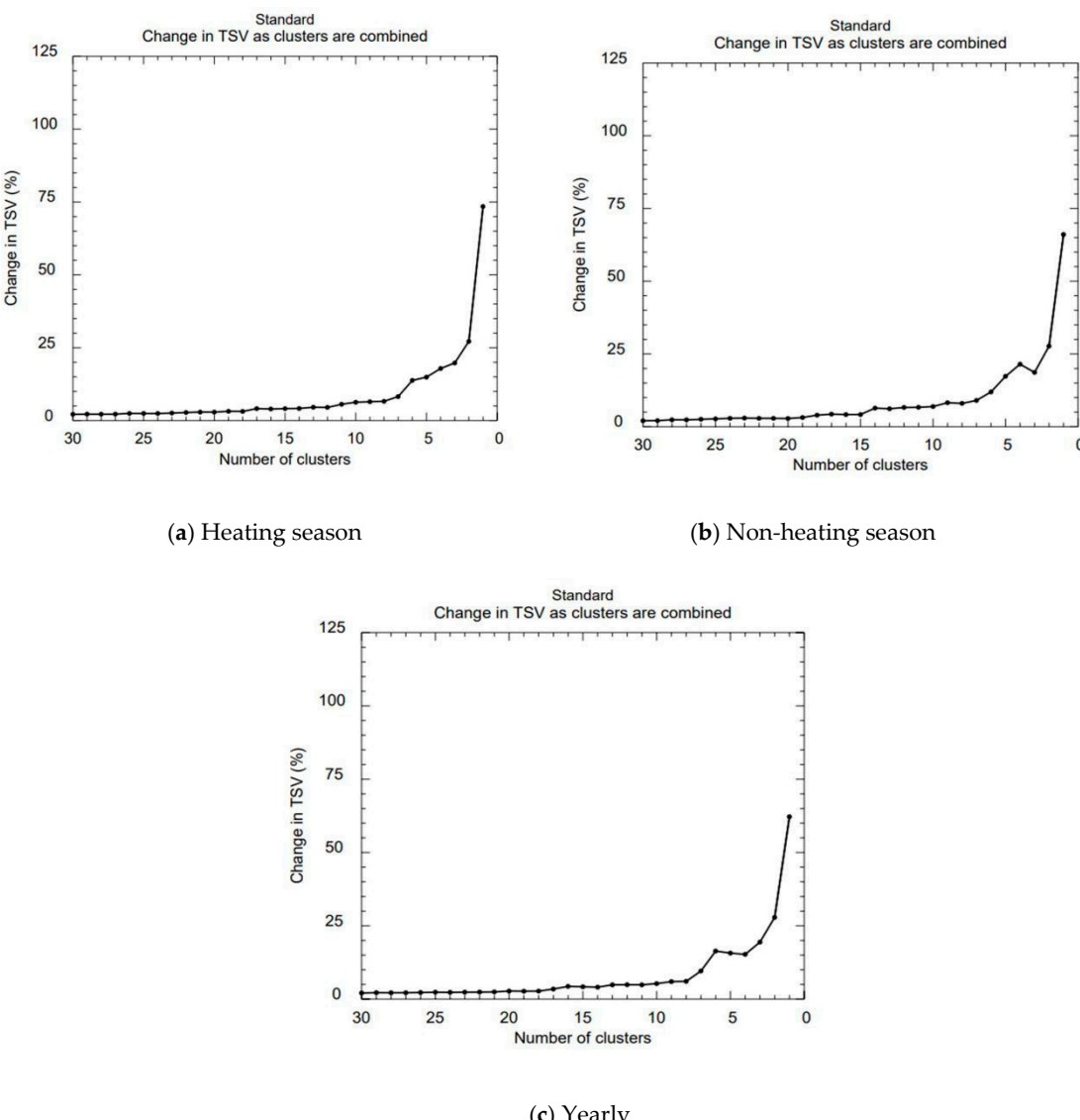

(**a**) Heating season      (**b**) Non-heating season

(**c**) Yearly

**Figure 6.** Changes in TSV (total spatial variance) as clusters are combined.

Heating Season

The trajectory clusters for the heating period with their 3-D characteristics (Figure 7) represents the central trajectories of the clusters with specific elevation and transportation profiles. Cluster 5 is the most common (37%) trajectory pathway in the heating season, and it moves slowly on lower altitudes to the northwest of the pollutant source region. It is the most critical and risk inducing pathway in regional zones due to its frequency and transport patterns. The most influenced areas are the northwest part of the Pavlodar region in Kazakhstan and the southern Russia, including Novosibirsk city. Cluster 1 and 2 also have low and slow transportation patterns, and they reach the highest elevation of 350 and 700 m, respectively. They are mainly distributed to the north and southeast directions resulting in the establishment of potential risks in the regions of southern Russia, the Republics of Altai and Khakassia, the Tuva Republic, and the central region of Kazakhstan,

including the capital city, Nur-Sultan. The fast and higher elevation pathways, Cluster 3, 4, and 6, transport to the east-northeast directions having an impact on the south and central lands of Russia, especially the Siberian regions. At the top elevation layer, which is typically higher than 2000 m, some trajectories are capable of reaching the south regions of Mongolia and the north-east lands of China, furtherly reaching the Yellow Sea on a long-term basis. If the global distribution is considered, the trajectories at higher elevation levels are also capable of penetrating Russia, passing through the Siberian region into the Arctic Ocean. It can be concluded that the air parcels in Clusters 1, 2, and 5 are likely to be riskier with increased levels of Hg concentrations than Clusters 3, 4, 6.

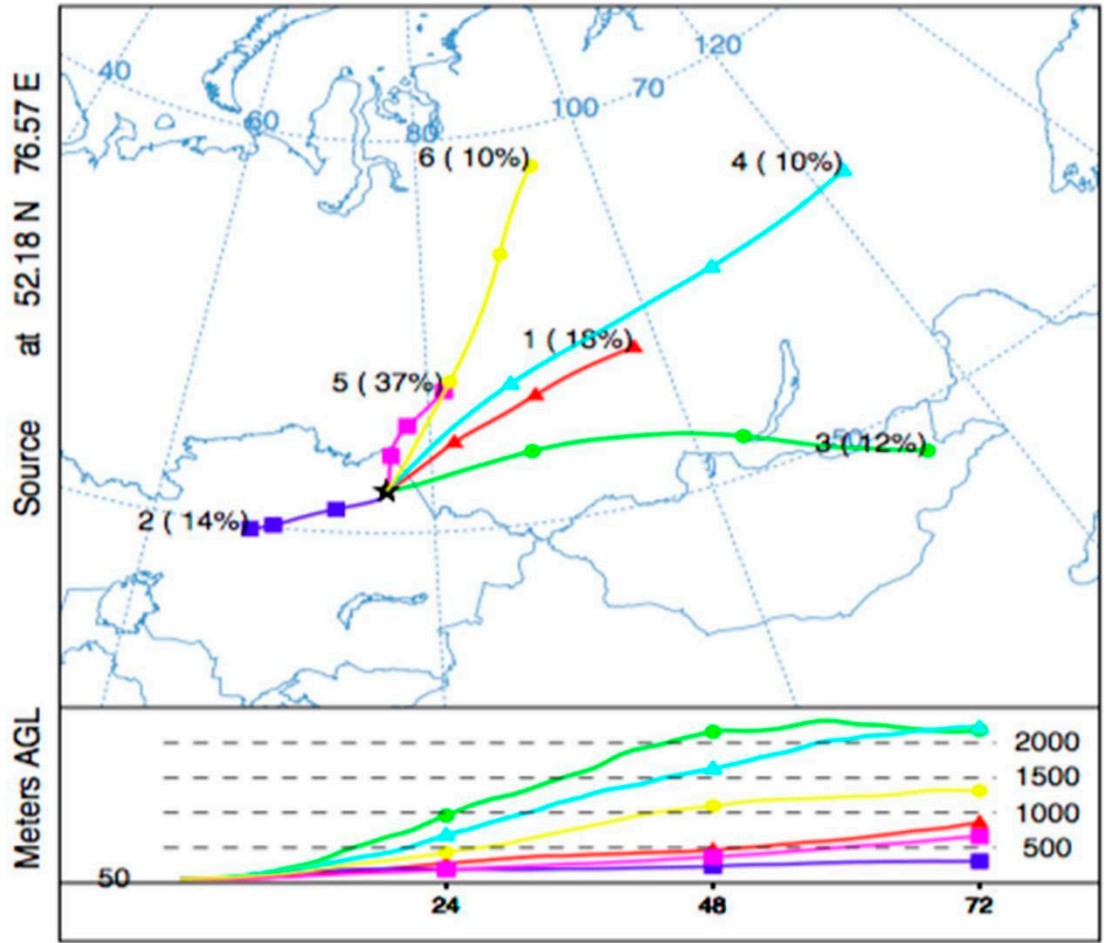

**Figure 7.** Optimum trajectory clusters (n = 6) for heating period.

Non-Heating Season

Figure 8 provides the central trajectories of the extracted clusters and their prevalence for the non-heating period. The slow and low trajectories dominate with 85% prevalence (Clusters 1, 3, and 4), which is significantly higher than heating season clusters. It can be suggested that the study area may have higher levels of impact and risk for the local and regional scales during the non-heating period. The environmental risks associated with the low and slow transport patterns of the dispersion routes might be folded up with a higher level of Hg evasion rates due to summertime atmospheric conditions. The source region has main distribution patterns, mainly widening in northwest and southeast directions. However, on the lower elevation levels (below 700 m), the trajectories with the highest air parcel densities are noted to have a potentially adverse effect on the lands of central Kazakhstan and the border with Russia, resulting in an increased probability of high concentrated parcels in those regions. The direction of the non-heating season clusters

does not differ that much from the heating season ones, but typically the impact distances are shorter in the non-heating period.

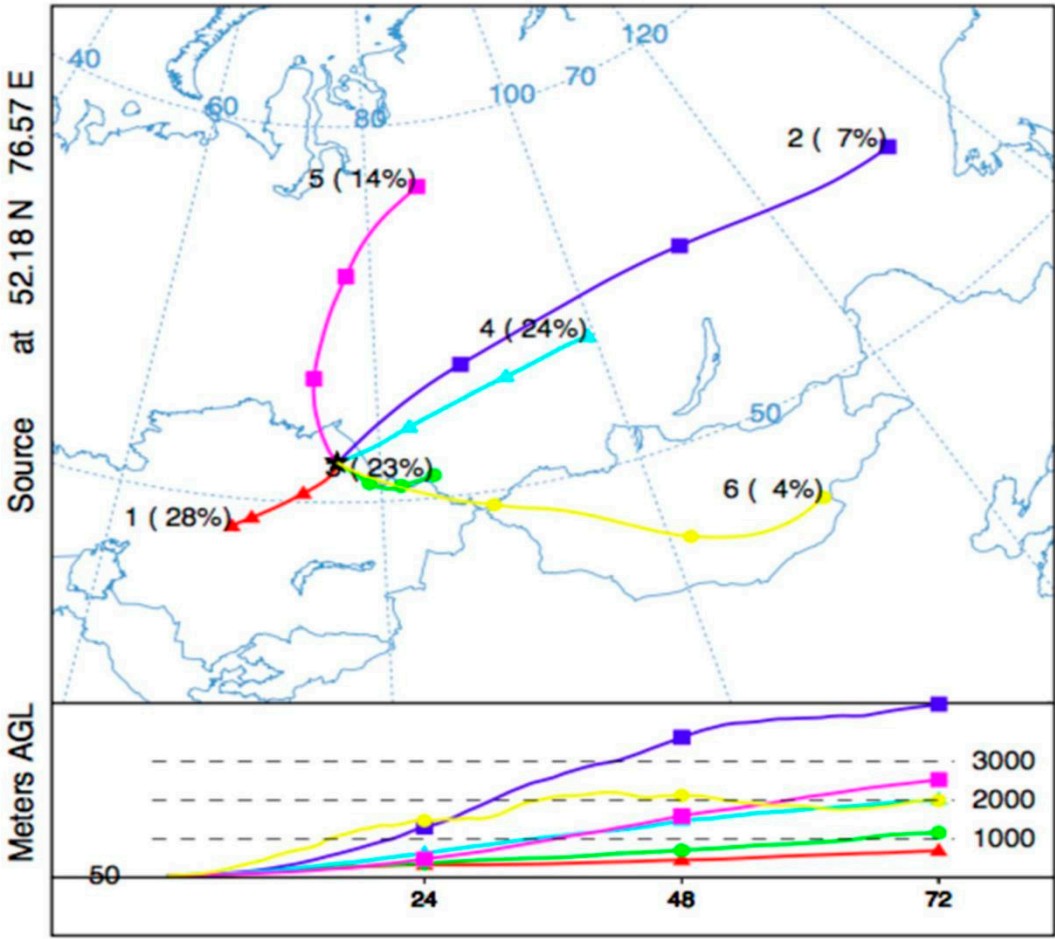

**Figure 8.** Optimum trajectory clusters (n = 6) for non-heating period.

## 4. Conclusions

The present study focused on the atmospheric deposition, emissions, and transport of mercury (Hg) in the region around a former Hg-cell chlor-alkali plant (CAP), including the area around Lake Balkyldak (a wastewater pond that received Hg from the CAP) as well as the nearby urban zone, the city of Pavlodar.

The atmospheric Hg levels measured in the study area were high around the territory of the CAP (16–22 ng/m$^3$) and its neighbor Lake Balkyldak (13–37 ng/m$^3$) compared to the measurements in the city during Campaign 1 (4–10 ng/m$^3$). Interpolation maps generated by using the kriging interpolation method and ArcGIS software support that the former CAP territory and Lake Balkyldak are the main sources of atmospheric Hg in the studied region. Longer and more extensive atmospheric Hg measurements in the Pavlodar region could be recommended to complete a full environmental assessment of the impact of CAP operations on the surrounding area. Moreover, several studies that investigated similar cases with the atmospheric impact of Hg-cell CAP operations detected Hg levels in the air within the plant's territory substantially greater than those measured outside of the territory. Therefore, atmospheric Hg concentrations inside the CAP territory in Pavlodar should be studied next.

The fugacity modeling may be beneficial in accurately predicting the environmental partition of contaminants as well as in choosing effective site remediation techniques. More advanced these models are, more details regarding a contaminant and its speciation as well

as site info are necessary for obtaining accurate results. The simpler Level I analysis used in the present study estimated almost the entirety of the Hg to escape to the atmosphere. The results obtained in the QWASI and HERMES models run for the lake system (based on the input Hg concentration in air, which were measured during the recent field trips) indicated a significant possible accumulation of Hg in lake sediments. The results suggest Lake Balkyldak a significant source of Hg. More information about the concentrations of Hg in different media (soil, sediments, water, and especially in air) inside the territory of the plant as well as in the city area, and a set of the local environmental parameters may further improve the results of the fugacity models.

The trajectory projections predicted the dispersion of atmospheric Hg over the whole territory of Mongolia, northwest China, and southwest Kazakhstan. The HYSPLIT-4 cluster analysis indicated the regions of southern Russia, the Republics of Altai and Khakassia, Tuva Republic, and the central region of Kazakhstan as the most susceptible regions due to low and slow transportation patterns of trajectory pathways during the heating season. High concentrations of Hg in the Siberian region have been documented in the literature; however, the verification of the pollutant's origin as the Pavlodar region has yet to be done.

The studied CAP ceased its activities more than 30 years ago. Furthermore, past remediation efforts cleaned up some of the Hg on the site. That being said, the performed fugacity analysis, the results of the performed site measurements for air Hg concentrations, and the results for long-term transport modeling of Hg indicate the presence of and potential human health risk from Hg originating from this site. Overall, the residual Hg pollution on the site seems significant and warrants a further investigation covering different environmental media: soils, sediments, and surface, and groundwater.

**Supplementary Materials:** The following are available online at https://www.mdpi.com/2073-4433/12/2/275/s1, Section S1: Detailed description of input parameters for fugacity models, Table S1: Semivariogram parameters and prediction error statistics, Table S2: Field measurement results of ambient TGM in Pavlodar, Table S3: Model compartment dimensions for Level I fugacity modeling, Table S4: Input parameters for QWASI model, Table S5: Input parameters for HERMES model, Figure S1: Atmospheric Hg dispersion over region around Lake Balkyldak during Campaign 2.

**Author Contributions:** Conceptualization, M.G., V.I. and F.K.; methodology, M.G., A.K., Z.A. and F.K.; software, A.K. and Z.A.; validation, M.G., A.K., Z.A., S.K., K.B. and F.K.; formal analysis, M.G., A.K., Z.A. and F.K.; investigation, M.G., A.K., Z.A., S.K., A.Z. and F.K.; resources, M.G., V.I. and F.K.; data curation, M.G., A.K., Z.A. and F.K.; writing—original draft preparation, M.G., A.K., Z.A., S.K. and K.B.; writing—review and editing, M.G., A.K., Z.A., S.K., K.B. and F.K.; visualization, Z.A. and A.K.; supervision, M.G., V.I. and F.K.; project administration, M.G. and F.K.; funding acquisition, M.G., V.I. and F.K. All authors have read and agreed to the published version of the manuscript.

**Funding:** The present research was supported by Nazarbayev University Competitive Grants Program (Funder Project Reference: 090118FD5319, Project Financial System Code: SOE 2018020).

**Institutional Review Board Statement:** Not applicable.

**Informed Consent Statement:** Not applicable.

**Data Availability Statement:** Data is contained within the article or Supplementary Materials.

**Conflicts of Interest:** The authors declare no conflict of interest.

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
