# Peer review of "Environmental Partitioning, Spatial Distribution, and Transport of Atmospheric Mercury (Hg) Originating from a Site of Former Chlor-Alkali Plant"

_atmosphere, doi:10.3390/atmos12020275_

Round 1
Reviewer 1 Report
Overall, I would like to congratulate the authors for the effort they have put in to improve the manuscript. It is substantially better and follows up much improved structure. After these improvements, I would recommend it for publication. There are two minor issues I would still recommend.
(1) I believe the back-trajectory component of the study is still far too over elaborative. This is a relatively minor source of mercury. The maximum concentration measured in this study is 37 ng/m3. There are many studies of near source Hg measurements that at >1000 ng/m3 (some >100000). Dilution of emissions from this single source will substantially diminish its effects on surrounding areas and particularly to long-range transport. I just feel this is too much information on something with such little real world impact.
(2) While the addition made in response to Review #1 comment #15 is good. Most of this is quite complimentary to the manuscript and could easily be put into the supplementary information. It doesn't add much value to the reader, but in the SI it is available if some readers want it.
Again, I appreciate the effort of the authors in improving this manuscript and wish them best of luck with their future research.
Reviewer 2 Report
The study of Guney et al. 2021 presents and analysis a short-term (from 23-26 July 2019) measurement of the gaseous elemental mercury (GEM) at a Hg-cell Chlor-alkali plant (CAP) in Pavlodar city in north-eastern Kazakhstan and try to understand the baseline concentration, behavior of the Hg release from CAP and the zone of its impact due to atmospheric transport. The topic of this study is interesting since the measurement carried at a major Hg emission source (CAP). However, I still feel that the data present in this study is very few, raising several concerns associated with the discussion and conclusion. Some of the figures are lost in this version. Following are my specific comments.
Lines 10-11. It should be better to claim that mercury is a toxic bioaccumulative trace pollutant, with the atmosphere being the important environment for global distribution.
Line 12: The traditional term is total gaseous mercury (TGM).
Line 14: “Hg-cell chlor-alkali plant” should appear in line 12 when you first-time mention CAP.
Line 20: encountered => observed?
Line 122: and 3) to evaluate the impact . . .
Lines 140-141: No need to imply further study in the method section.
Line 150: give a definition for TGM here.
Lines 154-159: Given that atmospheric dispersion strongly influences the Hg concentrations at the surrounding sites, could the authors comment on the general meteorological factors during the campaigns, thus we can expect the dispersion to somewhat similar among 4 campaigns.
Line 227: Is it GDAS-1 database?
Line 234: Add resolution of TGM data to this line.
Fig. 3 and 4: please state the unit in ng/m3 instead of ng/m^3. Also, the discussion associated with these figures are very few in the main text.
Line 273: decrease in contamination level compared to . . . ?
Lines 301-303: Given that GEM (the dominance fraction of TGM) is water insolubility, thus the effect of rainfall is insignificant. Therefore, the explanation here somewhat inconclusive.
Line 323: Please give more detail for refs 14 and 63. Which one represents China and Sweden.
Line 338: significant number should be 52.4 ng/L
Line 355: Clarify the measurement in 2018. Did the authors include the observation data in 2018 in this study?
Any data of Hg concentrations in sediment, water, vegetation, and flux measurements have been performed in this study area?
Line 382: This statement is speculative.
I didn’t see Fig. 5, 6, 7 and 8 in this version of the manuscript therefore I can’t give any detailed comments on the discussion in lines 426-530.
General comments:
Is it suitable to use 72-hour forward trajectories to determine the global impact? I think the authors can focus to discuss the impacted regions that suffer from the Hg emitted from CAP rather than quantitatively discuss the local-regional impacts.
In sections 3.3 and 3.4, the authors try to discuss the impacts of air masses originated from the CAP on local to global areas. However, all of the discussion and conclusions drawn in this part mainly based on the trajectory model without considering Hg emitted from the CAP. Without the inclusion of Hg concentrations in the air, it’s highly speculative to discuss the impacts and risks of contaminated Hg from CAP. In addition, given that Hg re-emission could be strongly seasonal dependent due to differences in ambient air temperature, then the impacts of outflow from CAP on surrounding areas should be varied seasonality and associated with the Hg concentrations. The present study only presents very short measurement results and cannot represent both heating and non-heating seasons. Therefore, I wonder how the authors can deal with these uncertainties?
Lines 476-480 More measurement studies are needed to demonstrate this hypothesis.
Line 532: Clarify “deposition” here. Atmospheric deposition or . . . ?
Lines 533: no need here since it repeated several times throughout the manuscript.
Lines 545-547: I don’t capture the main idea of this statement.
Lines 567-569: Is it from this study results?
Author Response
Please see the attachment.

This manuscript is a resubmission of an earlier submission. The following is a list of the peer review reports and author responses from that submission.
Round 1
Reviewer 1 Report
The authors present what they attempt to be a holistic study of Hg distribution around a CAP in Kazakhstan. Their modelling approach to do this does have some merit. However, I find it difficult to recommend the manuscript for publication for a number of reasons:
- The structure of the publication is very confusing, which makes it very hard to understand for the reader. This also makes it very difficult to interpret the key messages of the paper - that I am still not fully sure of. It is very strange to present environmental modelling based on empirical atmospheric measurements before the measurements themselves are presented.
- There is no description of the quality assurance and control of the LUMEX instrument used in this paper. Without those details how can we know the accuracy of the measured results; how was the instrument validated?
- There is a major assumption that Hg0 from this plant is being transported all across central asia based on elevated data measured up to about 10km from the CAP. Some more background assessments of this at sites a little further away in the background direction would have substantially benefitted the paper. There are huge amounts of (over-elaborative) discussion based on back trajectories and long-range transport that there is very little data to support. This does describe some of the experimental design short-comings of this this. Another aspect would have been taking measurements closer to the site (or even within the site) in the down wind direction. The closest measurements to the CAP in the downwind direction are 5-10km away on the other side of the lake, thus how can we know if there is a significant contribution (re-emissions) from the lake itself or just the CAP?
- I cannot find the input parameters for the size of the box model compartments used. This basically makes the model output concentrations meaningless.
- There is a lot of redundant and repetitive writing and information in the manuscript.
Thus, unfortunately I cannot recommend acceptance without a thorough streamlining and improvement to the current state of the manuscript.
Specific comments:
Line 45: "contributing" should be "augmenting" and this sentence has grammar issues.
Line 59-60: Sentence is redundant and should be deleted.
Lines 66-67: "and used Hg-cell technology to produce chlorine and caustic soda," redundant and should be deleted.
Lines 81-90: all basic knowledge and not necessary in the introduction. Delete.
Lines 102-105: overelaboration of unnecessary detail. Description in the methods is sufficient here.
Lines 134-142: how was this instrument quality controlled? Where there tests of any reference material? It is not simply OK to take measurements without assuring that those measurements are actually correct.
Line 195: The order of the results and discussion are very confusing. Modelling first based on data that hasn't even been described yet.
Lines 202-213: There concentration values in this paragraph are quite meaniningless. Where is this theoretical and very high concentration? Everywhere in the atmospheric compartment of the box model? The size of the compartments is not described. It also doesn't make sense because the atmospheric concentrations are the input values.
Lines 261-267: This should be included first. The measured data should also be included as a table in the SI.
Lines 268-278: This should be in the methods section.
Lines 292-296: The spatial distribution of the concentrations is bias by winds and the sampling. There are few measurements right near the CAP.
Lines 303-310: Here the authors touch upon other sources of Hg, but this is not well described in the study.
Lines 330-331: can a comment be made about how measurements in this study were not really that close in the down-wind direction.
Lines 371-377: so here are the authors suggesting a 0.5 increase in background Hg concentration from the plant 1500km is likely to cause major Hg inputs to this distant locations that will have significant health effects? Very speculative.
Lines 387-398: This has basically nothing to do with this study. If there is no attempt to quantify these inputs then it should not be discussed.
Lines 399-409: How is this different from the previous section.
Lines 410-441: Way too much overelaboration and repeated information.
Reviewer 2 Report
Table 2 needs reorganizing, especially for the HERMES inputs.
They are NOT correct, by not representing the column headings.
Some English grammar and syntax corrections needed.
Not enough mention on how the (CH3)2Hg is identified needs clarification to make partitioning claims valid.
Hysplit results need further, detailed explanation.